# Comparative Efficacy and Safety of JAK Inhibitors in the Management of Rheumatoid Arthritis: A Network Meta-Analysis

**DOI:** 10.3390/ph18020178

**Published:** 2025-01-28

**Authors:** Hani M. Almoallim, Mohammed A. Omair, Sameh A. Ahmed, Kota Vidyasagar, Bisher Sawaf, Mohamed A. Yassin

**Affiliations:** 1Department of Medicine, Faculty of Medicine, Umm Al-Qura University, Makkah 24382, Saudi Arabia; hmmoallim@uqu.edu.sa; 2Rheumatology Unit, Department of Medicine, King Saud University, Riyadh 11451, Saudi Arabia; 3Pharmacognosy and Pharmaceutical Chemistry Department, College of Pharmacy, Taibah University, Madinah 42353, Saudi Arabia; shassan@taibahu.edu.sa; 4Department of Pharmacy, University College of Pharmaceutical Sciences, Kakatiya University, Warangal 506001, India; 5Department of Internal Medicine, Toledo University Medical Center, Toledo, OH 43614, USA; 6Hematology Section, National Centre for Cancer Care and Research, Hamad Medical Corporation, Doha P.O. Box 3050, Qatar

**Keywords:** adverse drug reactions, JAK inhibitors, network meta-analysis, placebo, rheumatoid arthritis

## Abstract

Background/objective: Janus Kinase inhibitors (JAKinibs) are effective and well-tolerated targeted therapies for rheumatoid arthritis (RA). The comparative efficacy and safety of different JAKinibs remains unclear. This network meta-analysis (NMA) aimed to assess the relative efficacy and safety of different available JAKinibs. Methods: Searches were conducted on PubMed, CENTRAL, and ClinicalTrials.gov for randomized, double-blind, placebo-controlled trials comparing JAKinibs in RA patients. A frequentist NMA using the Netmeta package in R (R.4.3.0) was performed to evaluate both efficacy and safety outcomes. Continuous outcomes were presented as mean differences (MDs) and binary outcomes as relative risks (RR) with 95% confidence intervals (CI). The Cochrane risk of bias tool was used to assess the risk of bias in the included trials. Results: The analysis encompassed 39 trials with a total of 16,894 participants. Six JAKinibs (tofacitinib, baricitinib, upadacitinib, decernotinib, peficitinib, and filgotinib) were compared. Decernotinib at a dose of 300 mg showed a higher ACR50 response than other JAKinibs (RR = 7.55, 95% CI: 3.48 to 16.39, *p* < 0.01, surface under the cumulative ranking curve (SUCRA): 0.92). Tofacitinib at a dose of 1 mg twice daily had a significantly lower incidence of adverse drug reactions (ADRs) compared to other JAKinibs (RR = 0.80, 95% CI: 0.65 to 0.99, *p* = 0.04, SUCRA: 0.89), filgotinib 100 mg had a significantly lower infection risk (RR = 0.40, 95% CI: 0.21 to 0.79, *p* < 0.01, SUCRA: 0.90), whereas baricitinib 4 mg had the significantly highest herpes zoster risk (RR = 4.79, 95% CI: 1.03 to 22.21, *p* = 0.05, SUCRA: 0.11) compared to other JAKinibs. Conclusions: This NMA’s results indicate that commercially available JAKinibs show superior ACR responses and have comparable tolerability to placebo.

## 1. Introduction

Rheumatoid arthritis (RA) is a chronic autoimmune disease that affects about 1% of the world’s population, manifesting with a higher incidence in women, who are two to three times more prone to developing it compared to men. RA usually appears in individuals aged 30 to 60, with the frequency differing depending on the area and demographic. For instance, there are higher rates of occurrence in Northern Europe and North America when compared to Asian and African populations. This is attributed to genetic predispositions and environmental influences [1]. The disease is characterized by synovial membrane inflammation, leading to joint damage and consequential pain. The impact of RA on quality of life is significant, as it can lead to considerable disability, reduced mobility, and an increased risk of additional health complications [2,3]. The economic implications of RA are considerable; the Centers for Disease Control and Prevention reports that the healthcare costs related to RA in the United States exceed USD 19 billion annually [2]. Various RA management strategies exist, including disease-modifying antirheumatic drugs (DMARDs) and biologic agents [3]. These therapies have drawbacks such as suboptimal efficacy, safety concerns, and high costs. Some patients may not respond satisfactorily to these treatments or encounter adverse reactions [4]. Janus kinase inhibitors (JAKinibs) are a newer class of drugs that have shown effectiveness in the management of RA by focusing on essential intracellular signaling pathways for inflammation. Their advantages consist of notable decreases in joint inflammation, enhancements in physical function, and reductions in disease activity scores. JAKinibs have demonstrated effectiveness in patients who do not respond to traditional DMARDs and biologics, such as tumor necrosis factor inhibitors (TNFi), interleukin-6 receptor antagonists (IL-6 inhibitors), and B-cell depleting agents, offering an important option for challenging cases. Nevertheless, there are risks associated with the use of JAK inhibitors. Frequent negative reactions consist of infections in the upper respiratory tract, headache, and disturbances in the gastrointestinal system. Regulatory agencies advise careful use in high-risk populations due to reported serious consequences like higher chances of blood clotting events, raised cholesterol levels, and malignancies. Moreover, ongoing pharmacovigilance and careful patient monitoring are required to investigate the long-term safety of JAKinibs and balance risks with therapeutic benefits. The significance of personalized treatment and risk evaluation is emphasized when prescribing JAKinibs due to their dual effects [5].

Tumor necrosis factor-alpha (TNF-α) is essential is crucial in the progression of RA as it initiates inflammation leading to synovial swelling, cartilage damage, and bone erosion. Elevated levels of TNF-α in the synovial fluid and serum of patients with RA highlight its crucial role in promoting disease progression. TNFi functions by preventing TNF-α from attaching to its cellular receptors, decreasing the activation of inflammatory signaling pathways that play a key role in RA progression [6]. TNFi such as infliximab and adalimumab have had a major impact in treating RA by significantly reducing inflammation, joint damage, and overall disease activity. These biologics have also been associated with improved patient-reported outcomes, including enhanced physical function and quality of life. Despite their accomplishments, hurdles still exist, including the expensive nature of these treatments, which hinders access for numerous patients, particularly in areas with limited resources. Furthermore, as many as 30% of patients do not show a response to TNFi or experience a decrease in effectiveness over time, requiring different treatment methods. Additionally, there is still a worry about the potential for adverse events like severe infections, tuberculosis reactivation, and malignancies with prolonged use [6].

On the other hand, JAKinibs are a more recent type of targeted therapies that aim to overcome certain restrictions. JAKinibs provide a wider range of cytokine inhibition than TNF-α inhibitors by blocking intracellular signaling pathways controlled by Janus kinases, crucial for various pro-inflammatory cytokines. Administering them orally offers a more convenient option compared to injecting biologics, increasing patient compliance and contentment. Additionally, JAKinibs have shown effectiveness in patients who do not respond well to or cannot tolerate TNF-α inhibitors, providing hope for this difficult group of patients. Nevertheless, the risk of negative outcomes such as blood clots, infections, and high cholesterol levels highlights the significance of selecting and closely monitoring patients throughout the course of treatment. Despite these obstacles, JAKinibs remain an important part of the changing the RA treatment scene, fulfilling crucial needs and broadening treatment choices for patients [7].

Several JAKinibs, including tofacitinib 5 mg twice daily or 11 mg once daily (extended release), baricitinib (2 mg or 4 mg once daily), upadacitinib (15 mg once daily), peficitinib (100 mg or 150 mg once daily), and filgotinib (100 mg or 200 mg once daily), have received approval for human use. They have demonstrated better effectiveness in relieving symptoms and decreasing disease activity in RA patients despite the possible negative consequences [8]. The selection of RA treatment should consider patient-specific factors like disease severity, existing comorbidities, and drug tolerability. It is crucial to evaluate the potential risks and benefits of JAKinibs thoroughly before initiating therapy [9]. However, there is limited information comparing the relative efficacy, safety, and dosing of different JAKinibs. Network meta-analysis (NMA) is a type of meta-analysis that constructs a network of trials, each comparing two or more treatments [10]. It uses statistical models to combine evidence from multiple trials, providing valuable insights into the relative effectiveness of different treatments. Moreover, NMA allows for indirect comparisons among different JAKinibs and dosages that have not been directly compared in a single trial. An NMA is thus indispensable for an exhaustive evaluation of the available evidence [5,11].

The objective of this study was to execute an NMA to evaluate the relative efficacy and safety of various JAK inhibitors and their doses in the treatment of RA. This investigation aims to synthesize existing evidence comprehensively and to facilitate clinical decision-making concerning the administration of JAK inhibitors for RA. This NMA evaluated JAKinibs approved by the FDA and EMA for RA treatment: tofacitinib, baricitinib, upadacitinib, and filgotinib. It also includes trials of the investigational JAKinib decernotinib, which is not yet in clinical use.

## 2. Results

A total of 1442 citations were initially identified, and 271 duplicates were excluded; 1171 citations were eligible for FPS. A total of 418 citations were eligible for SPS, and their full texts were retrieved; of these, 381 studies did not meet eligibility criteria. Thirty-nine randomized double-blinded placebo-controlled trials were ultimately included in the analysis. The study selection process is presented in an adapted Preferred Reporting Items for Systematic Review and Meta-analysis guidelines flowchart in Figure 1.

### 2.1. Study Characteristics

This analysis includes a total of 39 trials with 16,894 participants. Among them, 3727 (22%) participants received tofacitinib, 1831 (11%) received baricitinib, 481 (3%) received decernotinib, 2075 (12%) received filgotinib, 1272 (8%) received peficitinib, 2212 (13%) received upadacitinib, and 4839 (29%) participants received a placebo.

The included trials comprised 7 (22%) that were single-nation and 32 (78%) that were multinational. Among the single-nation trials, six were conducted in Japan and one was conducted in the Republic of Moldova. Twenty-two (56%) were phase II trials and the remaining were phase III (Table 1). The number of participants randomized in the included studies ranged from 29 to 1307.

### 2.2. Subject Characteristics

Among the 16,894 participants, 13,302 (78.1%) were female. The mean age of the included participants ranged from 49.2 to 57.3 years (Table 1). Seventeen (44%), fourteen (36%), and three (8%) trials included participants who responded inadequately to disease-modifying anti-rheumatic drugs, methotrexate, and tumor necrosis factor inhibitors, respectively. Five trials (12%) included participants who were treatment-naïve and had received stable doses in previous therapies. Patients received tofacitinib in 11 (28%) trials, baricitinib and upadacitinib in 7 (18%) trials each, filgotinib in 6 (15%) trials, peficitinib in 5 (13%) trials, and decernotinib in 3 (7%) trials.

### 2.3. Risk of Bias Assessment

Twenty-four of the thirty-nine RCTs (62%) had insufficient information to judge whether the randomization, sequence generation, and allocation concealment were adequate. All the included RCTs (100%) had sufficient information on blinding of participants and personnel and blinding of outcome assessment. All RCTs (100%) reported having complete data for all included participants. There was no reporting bias for all the included RCTs (100%). A summary of the review authors’ judgments regarding each PRISMA risk of bias item for individual trials is given in Figure 2.

### 2.4. Networks

Each line joining two treatments in networks of clinical outcomes in RA management, such as ACR20, ACR50, ACR70, HAQ-D1, ADRs, serious ADRs, and discontinuation due to ADRs, represents a direct head-to-head comparison. Pairs of interventions without direct connections were compared indirectly through a common comparator.

### 2.5. Efficacy Outcomes

#### 2.5.1. ACR20 Response

Of the 39 studies, 38 were included in the ACR20 response analysis, encompassing 6663 patients in the intervention arms and 1710 in the placebo arms. This NMA demonstrated significant overall superiority in efficacy of JAKinibs over placebo for all interventions. Decernotinib had the highest ACR20 response (RR = 2.61, 95% CI: 1.70 to 4.03, *p* < 0.01, SUCRA: 0.92) and filgotinib had the lowest response (RR = 1.57, 95% CI: 1.25 to 1.97, *p* < 0.01, SUCRA: 0.24). For details on other interventions, refer to Table 2. Details of the subgroup analyses are presented in Appendix A.

In the dose-wise analysis, decernotinib 200 mg showed the highest response (RR = 2.92, 95% CI: 1.98 to 4.31, *p* < 0.01, SUCRA: 0.94). Filgotinib 30 mg was the least effective (RR = 0.86, 95% CI: 0.35 to 2.10, *p* = 0.74, SUCRA: 0.08), although this finding was not significant. For details on other interventions, refer to Figure 3 and Figure 4.

#### 2.5.2. ACR50 Response

Thirty-three studies were included in the ACR50 response analysis, with 8407 patients in the intervention arms and 3777 in the placebo arms. This NMA indicated significant overall efficacy of JAKinibs over placebo for all interventions. Decernotinib had the highest (RR = 5.28, 95% CI: 2.08 to 13.37, *p* < 0.01, SUCRA: 0.91) and filgotinib had the lowest ACR50 response (RR = 2.37, 95% CI: 1.48 to 3.80, *p* < 0.01, SUCRA: 0.38). For details on other interventions, refer Table 2. Details of the subgroup analyses are presented in Appendix A.

In the dose-wise analysis, decernotinib 300 mg showed the highest response (RR = 7.55, 95% CI: 3.48 to 16.39, *p* < 0.01, SUCRA: 0.92) and filgotinib 150 mg was the least effective (RR = 0.38, 95% CI: 0.02 to 8.60, *p* = 0.54, SUCRA: 0.11), although this finding was not significant. For details on other drug doses, refer to Appendix A.

#### 2.5.3. ACR70 Response

Thirty-one studies were included in the ACR70 response analysis, involving 8995 patients in the intervention arms and 4195 in the placebo arms. This NMA revealed significant overall efficacy of JAKinibs over placebo for all interventions. Decernotinib had the highest (RR = 6.46, 95% CI: 2.08 to 20.08, *p* < 0.01, SUCRA: 0.83) and filgotinib had the lowest ACR70 response (RR = 1.77, 95% CI: 1.49 to 2.10, *p* < 0.01, SUCRA: 0.18). For details on other interventions, refer to Table 3. Details of the subgroup analyses are presented in Appendix A.

In the dose-wise analysis, tofacitinib 60 mg showed the highest response (RR = 11.51, 95% CI: 5.09 to 25.99, *p* < 0.01, SUCRA: 0.92) and peficitinib 25 mg was the least effective (RR = 1.13, 95% CI: 0.51 to 2.51, *p* = 0.54, SUCRA: 0.77), although this finding was not significant. For details on other drug doses, refer to Appendix A.

#### 2.5.4. HAQ-DI Score

Thirty-five studies were included in the HAQ-DI score analysis, with 10,473 patients in the intervention arms and 4272 in the placebo arms. This NMA showed significant overall superiority in efficacy of JAKinibs over placebo for all interventions. Upadacitinib was associated with the highest drop in HAQ-DI score (MD = −0.31, 95% CI: −0.42 to −0.20, *p* < 0.01, SUCRA: 0.82) and tofacitinib had the lowest drop (MD = −0.19, 95% CI: −0.28 to −0.10, *p* < 0.01, SUCRA: 0.33). For details on other interventions, refer to Table 3. Details of the subgroup analyses are presented in Appendix A.

In the dose-wise analysis, decernotinib 300 mg showed the highest response (MD = −0.42, 95% CI: −0.83 to −0.02, *p* < 0.01, SUCRA: 0.77) and filgotinib 30 mg was the least effective (MD = 0.16, 95% CI: −0.40 to 0.72, *p* = 0.57, SUCRA: 0.12), although this finding was not significant. For details on other drug doses, refer to Appendix A.

#### 2.5.5. Incidence of Adverse Drug Reactions (ADRs)

Thirty-five studies were included in the analysis of ADR incidence, with 11,069 patients in the intervention arms and 4797 in the placebo arms. This NMA showed a non-significant overall increase in ADRs compared to placebo. Tofacitinib had the lowest incidence of ADRs (RR = 0.99, 95% CI: 0.82 to 1.19, *p* = 0.90, SUCRA: 0.89), although this finding was not significant, and upadacitinib had the most frequent ADRs (RR = 1.50, 95% CI: 1.16 to 1.92, *p* < 0.01, SUCRA: 0.07). For details on other interventions, refer to Table 4. Details of the subgroup analyses are presented in Appendix A.

In the dose-wise analysis, tofacitinib 2 mg showed the lowest incidence of ADRs, and this finding was significant (RR = 0.80, 95% CI: 0.65 to 0.99, *p* = 0.04, SUCRA: 0.89). Upadacitinib 36 mg was the least safe JAKinib (RR = 1.71, 95% CI: 1.26 to 2.30, *p* < 0.01, SUCRA: 0.14). For details on other drug doses, refer to Appendix A.

### 2.6. Incidence of Serious ADRs

The analysis of serious ADRs included 35 studies, with 11,069 patients in the intervention arms and 4797 in the placebo arms. This NMA showed a non-significant overall increase in serious ADRs with JAKinibs compared to placebo. Tofacitinib had the lowest incidence of serious ADRs (RR = 0.80, 95% CI: 0.52 to 1.23, *p* = 0.31, SUCRA: 0.78), although this finding was not significant, and upadacitinib showed the highest incidence (RR = 1.66, 95% CI: 1.08 to 2.54, *p* = 0.02, SUCRA: 0.09). For details on other interventions, refer to Table 4. Details of the subgroup analyses are presented in Appendix A.

For the dose-wise analysis, decernotinib 50 mg showed the lowest incidence of serious ADRs (RR = 0.20, 95% CI: 0.01 to 3.64, *p* = 0.27, SUCRA: 0.84), and baricitinib 7 mg was the least safe JAKinib (RR = 3.04, 95% CI: 0.25 to 37.07, *p* = 0.38, SUCRA: 0.21), although neither finding was significant. For details on other drug doses, refer to Appendix A.

### 2.7. Incidence of Infection and Other Risks

#### 2.7.1. Infection Risk

Out of 39 studies, 27 were included in the infection risk analysis. In total, 3201 patients were in the intervention arms and 2688 in the placebo arms. This NMA showed a non-significant overall increase in infection risk for all interventions compared to placebo. Decernotinib was associated with the lowest incidence of infection risk (RR = 0.63, 95% CI: 0.12 to 3.33, *p* = 0.59, SUCRA: 0.77) and peficitinib showed the highest (RR = 1.34, 95% CI: 0.91 to 1.98, *p* = 0.14, SUCRA: 0.26). Neither finding was significant. For details on other interventions, refer to Appendix A.

In the dose-wise analysis, filgotinib 100 mg had the significantly lowest infection risk compared to placebo (RR = 0.40, 95% CI: 0.21 to 0.79, *p* < 0.01, SUCRA: 0.90) and baricitinb (2 mg) showed the highest risk (RR = 1.27, 95% CI: 0.74 to 2.19, *p* = 0.39, SUCRA: 0.24). For details on other drug doses, refer to Appendix A.

#### 2.7.2. Herpes Zoster Risk

Sixteen studies were analyzed for herpes zoster risk, involving 4603 patients in the intervention arms and 1597 in the placebo arms. The NMA results indicated a non-significant overall increase in risk. Peficitinib had the lowest herpes zoster risk (RR = 0.55, 95% CI: 0.12 to 2.50, *p* = 0.44, SUCRA: 0.80), although this finding was not significant. Baricitinib showed the highest and significant increase in herpes zoster risk (RR = 4.46, 95% CI: 1.01 to 19.69, *p* = 0.05, SUCRA: 0.12). For details on other interventions, refer to Appendix A.

In the dose-wise analysis, upadacitinib (12 mg) had the lowest risk (RR = 0.39, 95% CI: 0.04 to 4.10, *p* = 0.44, SUCRA: 0.71) and baricitinib (4 mg) had the significantly highest risk (RR = 4.79, 95% CI: 1.03 to 22.21, *p* = 0.05, SUCRA: 0.11). For details on other drug doses, refer to Appendix A.

#### 2.7.3. Thrombotic Risk

Thirteen studies, with 4296 patients in the intervention arms and 1913 in the placebo arms, were reviewed for thrombotic risk. This NMA showed no significant overall increase in thrombotic risk. Tofacitinib had the lowest thrombotic risk (RR = 0.40, 95% CI: 0.06 to 2.56, *p* = 0.34, SUCRA: 0.82) and upadacitinib had the highest risk (RR = 2.34, 95% CI: 0.40 to 13.64, *p* = 0.34, SUCRA: 0.24), although both findings were not significant. For details on other interventions, refer to Appendix A.

In the dose-wise analysis, filgotinib (100 mg) had the lowest risk (RR = 0.23, 95% CI: 0.01 to 5.93, *p* = 0.37, SUCRA: 0.76) and upadacitinib (12 mg) had the highest risk (RR = 4.07, 95% CI: 0.19 to 88.31, *p* = 0.37, SUCRA: 0.25), although neither finding was significant. For details on other drug doses, refer to Appendix A.

#### 2.7.4. Malignancy Risk

Eleven studies, including 4348 patients in the intervention arms and 2140 in the placebo arms, were analyzed for malignancy risk. This NMA indicated a non-significant overall increase in cancer. Upadacitinib had the lowest malignancy risk (RR = 0.66, 95% CI: 0.13 to 3.28, *p* = 0.61, SUCRA: 0.64) and baricitinib had the highest (RR = 1.13, 95% CI: 0.28 to 4.49, *p* = 0.86, SUCRA: 0.34), although neither finding was significant. For details on other interventions, refer to Appendix A.

In the dose-wise analysis, upadacitinib (15 mg) had the lowest risk (RR = 0.64, 95% CI: 0.06 to 6.27, *p* = 0.70, SUCRA: 0.63). However, the same drug (12 mg) had the highest risk (RR = 2.02, 95% CI: 0.19 to 21.92, *p* = 0.56, SUCRA: 0.35). For details on other drug doses, refer to Appendix A.

#### 2.7.5. Major Adverse Cardiovascular Events (MACE) Risk

Eleven studies, involving 5854 patients in the intervention arms and 2667 in the placebo arms, were included for MACE risk analysis. This NMA showed a non-significant overall increase. Upadacitinib had the lowest MACE risk (RR = 0.40, 95% CI: 0.07 to 2.43, *p* = 0.32, SUCRA: 0.68) and baricitinib showed the highest risk (RR = 0.97, 95% CI: 0.24 to 3.86, *p* = 0.96, SUCRA: 0.35), although neither finding was significant. For details on other interventions, refer to Appendix A.

For the dose-wise analysis, upadacitinib (15 mg) had the lowest risk (RR = 0.46, 95% CI: 0.08 to 2.74, *p* = 0.39, SUCRA: 0.67) and decernotinib (100 mg) the highest risk (RR = 4.00, 95% CI: 0.19 to 86.05, *p* = 0.38, SUCRA: 0.21), although neither finding was significant. For details on other drug doses, refer to Appendix A.

#### 2.7.6. Hypercholesterolemia Risk

Eleven studies, with 3252 patients in the intervention arms and 1443 in the placebo arms, were reviewed for hypercholesterolemia risk. This NMA showed a non-significant overall increase in hypercholesterolemia. Baricitinib had the lowest hypercholesterolemia risk (RR = 1.78, 95% CI: 0.97 to 3.27, *p* = 0.06, SUCRA: 0.58) and filgotinib the highest risk (RR = 4.93, 95% CI: 0.30 to 81.94, *p* = 0.27, SUCRA: 0.31), although both findings were insignificant. For details on other interventions, refer to Appendix A.

In the dose-wise analysis, tofacitinib (60 mg) had the lowest risk (RR = 0.64, 95% CI: 0.03 to 15.60, *p* = 0.78, SUCRA: 0.78) and filgotinib (200 mg) the highest risk (RR = 9.11, 95% CI: 0.53 to 155.22, *p* = 0.13, SUCRA: 0.23), although neither finding was significant. For details on other drug doses, refer to Appendix A.

### 2.8. Discontinuation Due to ADRs

Of the 39 studies, 34 were included in the analysis of discontinuation due to ADRs. There were 10,789 patients in the intervention arms and 4711 in the placebo arms. The NMA showed a non-significant overall increase in discontinuation rates compared to placebo. Peficitinib had the lowest incidence of discontinuation (RR = 0.74, 95% CI: 0.46 to 1.19, *p* = 0.21, SUCRA: 0.90) and decernotinib had the highest (RR = 1.63, 95% CI: 0.38 to 6.96, *p* = 0.51, SUCRA: 0.32), although neither finding was significant. For details on other interventions, refer to Table 5.

In the dose-wise analysis, decernotinib 50 mg showed the lowest incidence of discontinuation due to ADRs (RR = 0.20, 95% CI: 0.01 to 4.04, *p* = 0.29, SUCRA: 0.87). Tofacitinib 60 mg was associated with the highest incidence of discontinuation (RR = 3.32, 95% CI: 1.39 to 7.96, *p* < 0.01, SUCRA: 0.10). For details on other drug doses, refer to Appendix A.

### 2.9. Incidence of Death

Twelve studies were included in the incidence of death analysis, with 5052 patients in the intervention arms and 2637 in the placebo arms. This NMA showed a non-significant overall increase in death rates compared to placebo. Upadacitinib had the lowest incidence of death (RR = 0.33, 95% CI: 0.01 to 8.17, *p* = 0.50, SUCRA: 0.78) and filgotinib had the highest (RR = 9.0, 95% CI: 0.49 to 166.64, *p* = 0.14, SUCRA: 0.16), although neither finding was significant. For details on other interventions, refer to Table 5.

In the dose-wise analysis, upadacitinib 15 mg had the lowest incidence of death (RR = 0.33, 95% CI: 0.01 to 8.17, *p* = 0.50, SUCRA: 0.74). Filgotinib 200 mg had the highest incidence of death (RR = 7.00, 95% CI: 0.36 to 135.10, *p* = 0.20, SUCRA: 0.21), although this finding was not significant. For details on other drug doses, refer to Appendix A.

### 2.10. Publication Bias Assessment

Egger’s and Begg’s tests indicated no statistically significant publication bias for ACR20, ACR50, ACR70, HAQ-DI, discontinuation rate, death, infection risk, herpes zoster risk, thrombotic risk, malignancy risk, MACE risk, and hypercholesterolemia estimates (Egger test: *p* = 0.06, 0.6, 0.17, 0.82, 0.99, 0.7, 0.63, 0.40, 0.28, 0.88, 0.44, and 0.57; Begg’s test: *p* = 0.07, 0.15, 0.06, 0.87, 0.06, 0.2, 0.44, 0.65, 0.02, 0.48, 0.44, and 0.57, respectively; see Appendix A). However, a significant publication bias was indicated for ADRs and serious ADRs (Egger test: *p* = 0.0001 and 0.02; Begg’s test: *p* = 0.68 and 0.01, respectively).

## 3. Discussion

This NMA integrates the current evidence regarding the safety and efficacy of JAKinibs for patients with RA, based on data from double-blind randomized controlled trials. The analysis represents the most comprehensive and up-to-date review, encompassing 39 trials involving a total of 16,894 participants. By synthesizing these findings, this NMA enhances the understanding of JAKinibs in the management of RA and provides valuable insights that can better inform public health practices and clinical decision-making. RA is a chronic autoimmune condition characterized by systemic and progressive inflammation of the joints due to immune system dysregulation. The JAK/STAT signaling pathway has been implicated in inflammatory and autoimmune diseases, including RA, psoriasis, and inflammatory bowel disease [11]. Based on this knowledge, JAKinibs were developed with varying affinity for each receptor. Currently, the FDA and EMA have approved specific dosages for adults with moderate to severe RA who have not responded well to methotrexate: 5 mg of tofacitinib taken twice daily, 4 mg of baricitinib, 15 mg of upadacitinib, and 200 mg of filgotinib taken daily [8,50]. However, the lack of direct comparisons among different JAKinibs and the absence of comprehensive meta-analyses and NMAs evaluating the effectiveness and safety of the various dosages leaves a gap in evidence for definitive conclusions about the benefits and risks of each JAKinib and its dosages. However, more research is necessary to compare JAKinibs directly, including different dosages, to better understand how effective and safe they are in comparison to each other. Extended monitoring is essential for evaluating the long-lasting advantages and possible drawbacks linked to these therapies, especially in individuals with different underlying conditions and levels of disease severity. The effects of JAKinibs on chronic diseases like heart disease, diabetes, and cancer are not well studied and require thorough investigation. The presence of other health conditions can make treatment choices more difficult, requiring careful monitoring of how they interact with JAKinibs over time. Additionally, including information from patients about their outcomes, such as better pain, fatigue, and overall quality of life, will provide a more thorough insight into how these treatments enhance not only clinical symptoms, but also patient welfare. Apart from clinical effectiveness, other factors like adherence to treatment, use of healthcare services, and cost-effectiveness should also be considered in practical situations. These factors are essential in assessing the actual value of JAKinibs in managing RA and should be included in upcoming research to guide informed clinical and policy choices that enhance patient care.

Our analyses showed that JAKinibs significantly outperformed placebos in achieving ACR20, ACR50, and ACR70 responses. Notably, decernotinib was the most effective, followed by tofacitinib, while filgotinib was the least effective. These results align with those of a previous meta-analysis [7]. Looking at individual doses, decernotinib 200 mg (SUCRA: 0.94), decernotinib 300 mg (SUCRA: 0.92), and tofacitinib 60 mg (SUCRA: 0.92) demonstrated the highest ACR20, ACR50, and ACR70 responses, respectively. This finding contrasts with a previous NMA [7], which indicated that tofacitinib 5 mg had the second-highest ACR20 response. The discrepancy may be due to that NMA’s inclusion of only 11 studies and three types of JAKinibs (upadacitinib, baricitinib, and tofacitinib) [51]. In addition, our study highlights the importance of the dosage in the effectiveness of JAKinibs, showing that higher doses of decernotinib and tofacitinib result in better results, which is crucial for patients with severe rheumatoid arthritis. This implies that adjusting the dose carefully may be necessary to achieve the best therapeutic outcomes, especially in patients who are hard to treat. It is interesting that decernotinib was found to be more effective even at higher doses, indicating that certain JAKinibs may have increased potency at higher doses, potentially influencing treatment choices. Patient tolerance should be considered when adjusting doses, as higher doses could heighten the likelihood of adverse effects. In the end, tailoring treatment based on disease severity and individual patient traits is likely the best way to improve treatment results with JAKinibs [51].

JAK inhibitors also demonstrated significant efficacy in improving HAQ-DI scores compared to placebo, with upadacitinib (SUCRA: 0.82) showing the highest effectiveness and tofacitinib (SUCRA: 0.33) the lowest. This contrasts with findings from a previous meta-analysis [7]. The discrepancy may stem from differences in the number of included trials and the sample size, which could influence the robustness of the conclusions. Furthermore, the notable enhancement in HAQ-DI ratings, especially seen with upadacitinib, highlights the capacity of JAKinibs not only to enhance disease activity but also functional impairment, a crucial factor affecting quality of life for RA patients. This emphasizes the importance of JAKinibs in treating both the inflammatory and functional aspects of RA, leading to enhanced overall patient results. Adding functional improvement to clinical decision-making can assist clinicians in choosing the most suitable JAK inhibitor for patients, as a patient’s daily life is often influenced by their functional ability. Although RA disease activity is important, functional disability significantly impacts long-term outcomes, such as independence and overall well-being. Future research could gain advantage by examining the lasting impacts of JAKinibs on HAQ-DI scores, particularly as disease-modifying therapies advance [11].

Concerning safety, upadacitinib was the only JAKinib significantly associated with increased adverse events (AEs) and serious adverse events (SAEs), consistent with previous findings [7]. However, data on decernotinib and peficitinib are scarce, with five or fewer trials for each, limiting confidence in these results. Additionally, the short duration of trials for these drugs does not allow for conclusions on long-term safety. Ongoing phase 3 trials for decernotinib and peficitinib and the limited treatment duration observed to date underscore the need for further data to fully assess efficacy and safety. The occurrence of AEs with upadacitinib underlines the importance of personalized treatment strategies, particularly for patients with other health conditions that could worsen these dangers. In clinical practice, it is crucial to closely observe patients for new side effects, especially with recent drugs like decernotinib and peficitinib, until more safety data are gathered. Although upadacitinib is a valuable choice for numerous RA patients, its link to more adverse events means that caution is needed for patients with prior infections or heart problems. Additionally, it is essential to provide patient education, educating them about the possible side effects and the significance of timely identification and reporting of negative reactions. This will guarantee that in every case, the advantages of treatment are greater than the dangers, enabling a more tailored and secure approach to managing RA [51].

The most frequently reported adverse events with JAK inhibitor treatment in RA patients are infections [52]. While the incidence of common infections such as upper respiratory tract, lower respiratory tract, and urinary tract infections is higher compared to the general population, it is similar across different JAK inhibitors. Our results suggest that JAK inhibitors non-significantly increase the risk of infection. These findings are consistent with previously published evidence-based review [5]. Nevertheless, extra caution is needed for RA patients taking JAKinibs due to a higher risk of infection, especially for those who are older or have weakened immune systems. In medical environments, preventive actions such as vaccinations and regular screenings for infections should be taken to reduce this risk. Patients prescribed JAKinibs must be watched for signs of infection, with the potential need to modify or pause treatment if serious infections occur [5]. Additionally, having more comprehensive data on infection risks, including trends over a long period of time, would help clinicians determine the overall benefit–risk ratio for individual patients. Creating specific guidelines or protocols for handling infections in RA patients on JAKinibs could also be helpful, considering their higher vulnerability [5].

Regarding the risk of malignancy, our analysis suggests a non-significant increase. These findings are consistent with previously published meta-analyses, which have indicated no significant increase in the risk of malignancies in patients treated with either tofacitinib or DMARDs compared to those treated with conventional DMARDs or placebo [53]. To our knowledge, this is the first NMA that includes all JAKinibs and their doses from phase 2 and 3 double-blind placebo-controlled trials, facilitating drug and dosage comparisons across patient groups. We employed a mixed-treatment comparison meta-analytical method to provide a detailed understanding of clinically significant outcomes. Although the potential for cancer is still a worry when using JAKinibs for an extended period, the limited negative results in this study indicate that the dangers linked to JAKinibs are quite controllable, especially when compared to other immunosuppressive treatments. Nonetheless, due to the possible seriousness of cancer, continuous monitoring for detection in the early stages is essential in managing RA with JAK inhibitors [53]. Additional research investigating the malignancies linked to JAKinibs, along with their mechanisms, could be useful in pinpointing populations at risk. Furthermore, studying the relationship between JAKinibs and genetic predispositions to cancer may offer more individualized risk evaluations for patients with RA. These initiatives would assist in understanding the extended safety record of JAKinibs and enable better-informed clinical choices.

Finally, the subgroup analysis based on the quality of included studies highlights important differences in the efficacy and safety profiles of JAKinibs. Studies with a low risk of bias demonstrate consistent and significant efficacy across outcomes, particularly for ACR20, with manageable heterogeneity. In contrast, high-risk studies show larger relative effects for some outcomes, such as ACR50 and ACR70, but with increased variability and heterogeneity. These findings underscore the influence of study quality on the observed magnitude and reliability of treatment effects, emphasizing the need for robust methodological designs to derive accurate and clinically meaningful conclusions.

However, this study has limitations, including a small number of studies for some JAKinibs, potentially leading to overestimated outcomes for certain doses. Our NMA relied on journal articles and ClinicalTrials.gov, without satisfying the transitivity assumption due to heterogeneity in patient populations across studies; for instance, some studies included patients who were unresponsive to MTX treatment, while others recruited individuals unresponsive to DMARDs or who were treatment-naïve. Additionally, the duration of the included studies was relatively short, and data pertaining to the long-term effects of JAKinibs were lacking.

## 4. Materials and Methods

This NMA adhered to the Preferred Reporting Items for Systematic Reviews and Meta-Analyses (PRISMA) guidelines, particularly the extension for NMAs [54] as delineated in Appendix A. This protocol was registered with the International Platform of Registered Systematic Review and Meta-Analysis Protocols (INPLASY) (registration number INPLASY2024100084).

### 4.1. Eligibility Criteria

Our research question was structured using the PICOS format (Population, Intervention, Comparison, Outcome, Study Design) to determine study eligibility. We incorporated both published and unpublished phase II and III randomized double-blind placebo-controlled trials that assessed RA patients, without restrictions on age or sex. Studies that compared any JAK inhibitor against other JAK inhibitors or a placebo in the context of RA management were considered. Exclusions were made for open-label, single-blind, non-randomized controlled trials, quasi-experimental studies, observational studies, animal studies, case reports, reviews, editorials, abstracts, and any trials not published in English.

### 4.2. Search Methods

An exhaustive search of the literature was conducted across several databases, including PubMed, and The Cochrane Central Register of Controlled Trials (CENTRAL), for pertinent RCTs from their start dates up to September 2024. Search phrases such as “Rheumatoid arthritis” alongside names of specific JAK inhibitors and “RCTs” were employed. The search strategies are detailed in Appendix A. To identify additional published and unpublished trials, we manually examined reference lists from review articles, entries from ClinicalTrials.gov, the World Health Organization International Clinical Trials Registry Platform, and relevant bibliographies. No restrictions on date or language were imposed during the electronic searches.

### 4.3. Study Selection

Two authors independently performed first-pass screening (FPS) by reviewing the titles and abstracts of all the records retrieved to identify articles that potentially met the predefined eligibility criteria. The full texts of eligible titles were downloaded and reviewed independently by the two authors in the second-pass screening (SPS) to determine relevant inclusion in the final analysis. The discrepancies between the two reviewers during the FPS and SPS were sorted by discussion with a third reviewer.

### 4.4. Data Extraction and Management

Two authors independently extracted data from the included RCTs using data extraction templates. Discrepancies during the data extraction were resolved through discussion with a third reviewer. The following details were extracted: study identification, authors’ details, study objectives, study design, the setting of intervention, study population (including), measures, and main findings American College of Rheumatology 20% ACR 20, ACR50, ACR70, Health Assessment Questionnaire-Disability Index (HAQ-DI), ADRs, patients with serious ADRs, and patients discontinued due to ADRs.

### 4.5. Assessment of Risk of Bias in Included Studies

Two reviewers independently assessed the methodological quality of each study using the Cochrane Collaboration tool for assessing the risk of bias [55]. The following potential domains were assessed: random sequence generation; allocation concealment; blinding of participants and personnel; blinding of outcome assessment; incomplete outcome data; selective reporting; and other sources of bias. For each domain, the risk of bias was scored as low, unclear, or high.

### 4.6. Statistical Analysis

All statistical analyses were performed using the R programming language, version 4.3.0, to conduct a frequentist network meta-analysis (NMA). The primary outcomes of interest in this analysis were the responses measured by ACR20, ACR50, and ACR70, which assess 20%, 50%, and 70% improvement in symptoms, respectively. The secondary outcomes included the mean change in the HAQ-D1 score, which evaluates changes in disability, as well as the incidence of adverse drug reactions (ADRs), serious ADRs, and discontinuations due to ADRs, across both treatment and comparator groups. The NMA approach was particularly valuable for comparing various JAKinib treatment strategies and their different doses (we used the daily drug dose as the basis for comparing different doses of JAKinibs within the same drug class to standardize dosing regimens across studies), even when direct head-to-head comparisons were limited. Treatments were ranked based on the surface under the cumulative ranking curve (SUCRA). The relative risk (RR) was utilized to describe binary outcome variables, such as the number of patients responding to ACR20, ACR50, ACR70, and the number of patients experiencing ADRs, serious ADRs (as deemed serious by the corresponding author), and those who discontinued treatment due to ADRs. In contrast, continuous outcomes, such as the mean change in HAQ-D1 score, were reported as mean differences (MD) with 95% confidence intervals (CIs). To visually represent the mean effect size and confidence intervals for individual studies, estimates were displayed graphically in forest plots.

The magnitude of heterogeneity between the studies was assessed using the I2 statistic (% residual variation due to heterogeneity), and Tau2 (method of moments estimate of between-study variance) was used for each of the pooled estimates. I2 values range between 0 and 100% and are considered low for I2 < 25%, modest for 25–50%, and large for >50% [56]. As differences between the studies were very high (95–99% inconsistency), a random effect DerSimonian–Laird model was used in all analyses [56]. The risk of publication bias was inspected using the symmetry of funnel plots, as well as Egger’s and Begg’s tests. Subgroup analysis was performed for the confounding factors such as the type of population included (drug-naive/stable dose on previous therapy, inadequate response to DMRDs, inadequate response to MTX, and inadequate response to TNFi), duration of the treatment (up to 6 weeks, 12 weeks, and 24 weeks), and quality of included studies (low risk and high risk based on Cochrane RoB).

## 5. Conclusions

This NMA offers valuable information on the effectiveness and safety comparison of JAKinibs in RA. This research, based on data from 39 randomized controlled trials with 16,894 participants, demonstrates that JAKinibs are much more effective than a placebo in meeting important clinical goals like ACR20, ACR50, and ACR70 responses, and enhancing functional disability (HAQ-DI scores). Decernotinib was found to have the highest efficacy for ACR responses out of all the JAKinibs examined, whereas filgotinib exhibited relatively lower efficacy. Regarding safety, although most JAKinibs had tolerability like placebo, upadacitinib had a greater occurrence of adverse and serious adverse events. This highlights the importance of personalized treatment plans to improve therapy results and reduce potential risks. Moreover, the extensive range of cytokine suppression provided by JAKinibs, along with their oral dosing, marks a notable progression in treating RA, especially for individuals who do not respond to conventional DMARDs or TNF-α inhibitors. However, continuous pharmacovigilance is essential for the long-term evaluation of JAKinibs in RA treatment.

## Figures and Tables

**Figure 1 pharmaceuticals-18-00178-f001:**
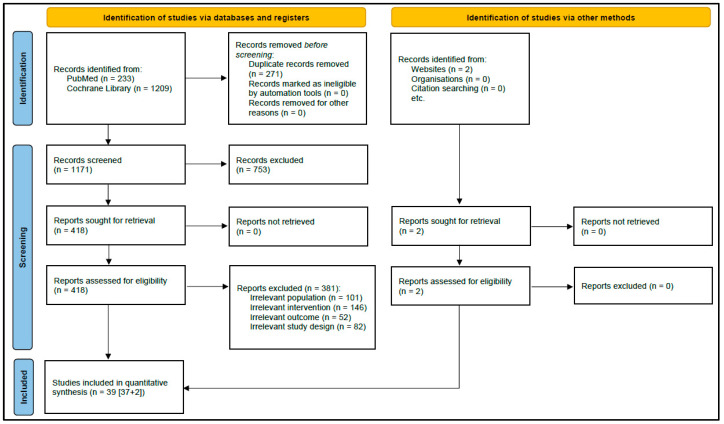
PRISMA flow diagram for study selection.

**Figure 2 pharmaceuticals-18-00178-f002:**
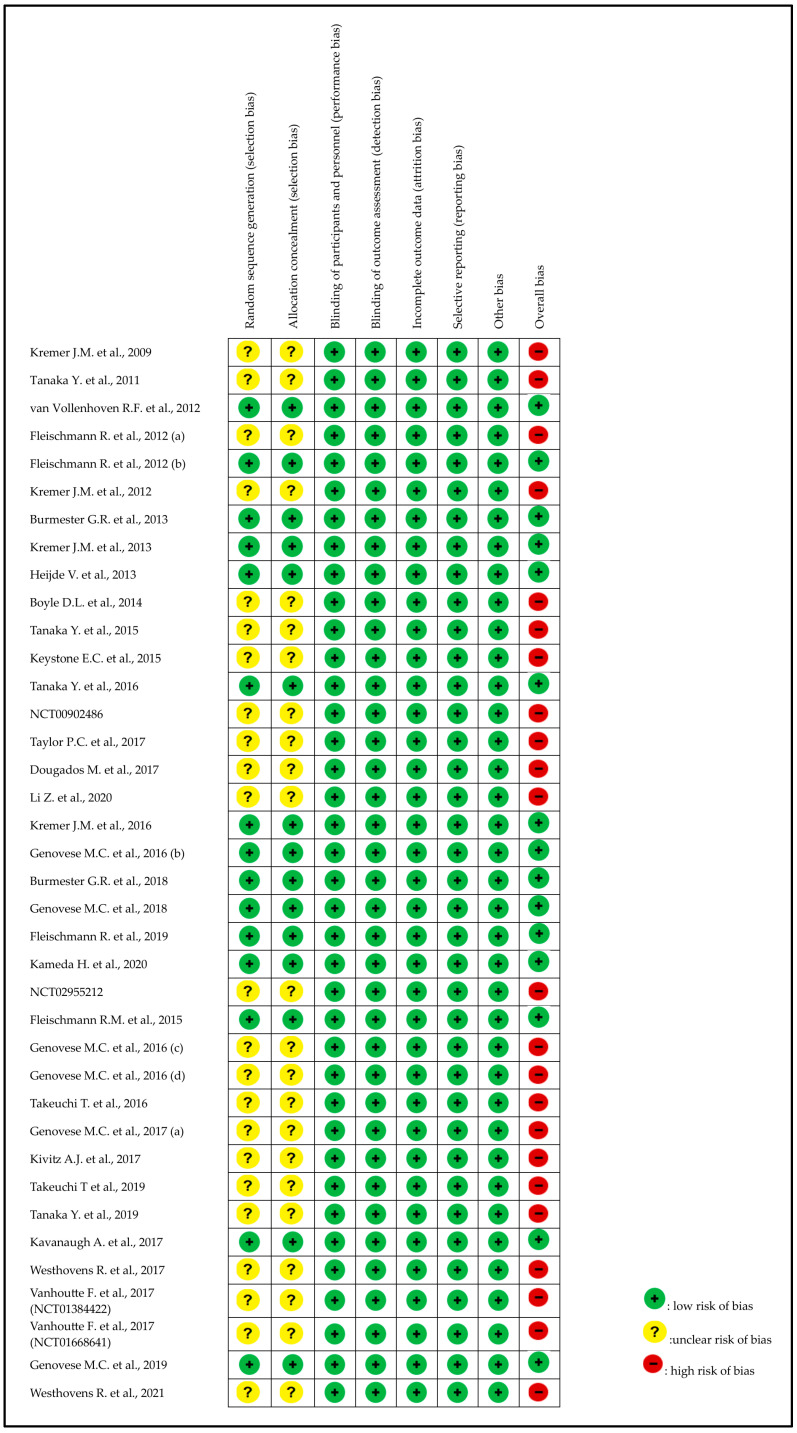
Assessment of the risk of bias in included studies with Cochrane domain-based quality assessment tool [12,13,14,15,16,17,18,19,20,21,22,23,24,25,26,27,28,29,30,31,32,33,34,35,36,37,38,39,40,41,42,43,44,45,46,47,48,49].

**Figure 3 pharmaceuticals-18-00178-f003:**
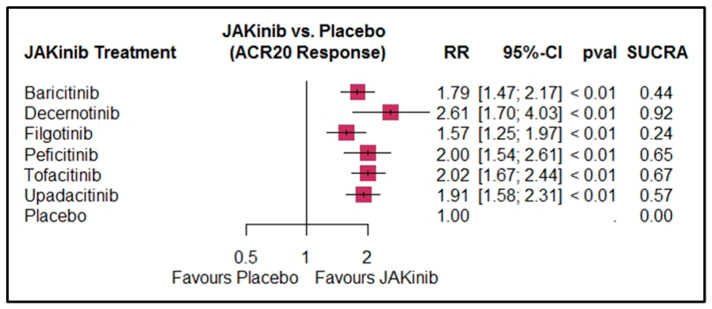
Forest plot for ACR20 response by JAKinib treatment. The size of the boxes represents the sample size/weight of the respective treatment groups, while the horizontal lines indicate the 95% CI.

**Figure 4 pharmaceuticals-18-00178-f004:**
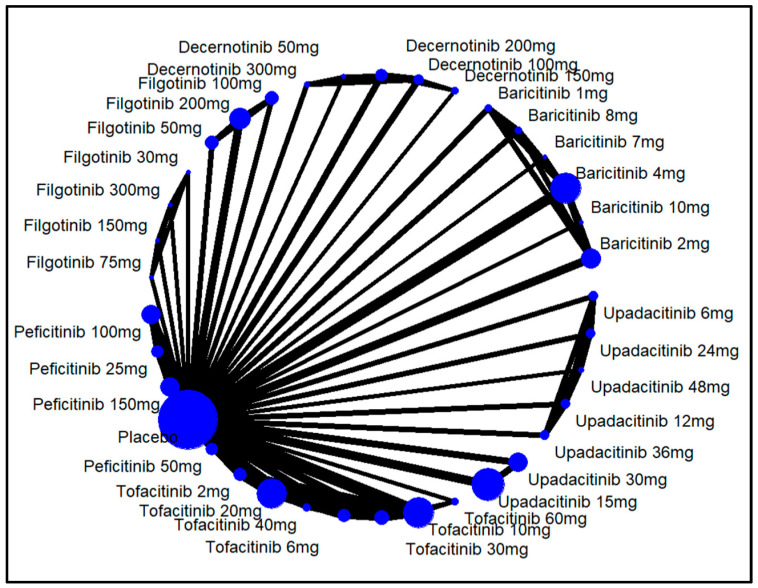
Network structure for ACR20 response by different doses of JAKinibs. The size of the blue circle represents the number of patients in each treatment group (larger circles indicate more patients). The thickness of the lines between the circles corresponds to the number of studies comparing the treatments directly (thicker lines indicate more studies).

**Table 1 pharmaceuticals-18-00178-t001:** Characteristics of the studies included in the network meta-analysis comparison.

Author Name	Trial Registration Number	Country	Duration of Disease	Number of Groups	Number Randomized	Mean Age (Years)	Sex (Male%)	Race (%)	Intervention (mg)
Kremer J.M. et al., 2009 [12]	NCT00147498	Multi	9.6	4	264	50.5	56.5	White > 50%	Tofacitinib *−5, 15, 30
Tanaka Y. et al., 2011 [13]	NCT00603512	Japan	7.64	5	140	51.3	86.18		Tofacitinib *−1, 3, 5, 10
van Vollenhoven R.F. et al., 2012 [14]	NCT00853385	Multi	7.8	3	717	53	20	White > 50%	Tofacitinib *−5, 10
Fleischmann R. et al., 2012 (a) [15]	NCT00550446	Multi	9	6	386	53	14	White > 50%	Tofacitinib *−1, 3, 5, 10, 15
Fleischmann R. et al., 2012 (b) [16]	NCT00814307	Multi	8.1	3	611	52.1	14	White > 50%	Tofacitinib *−5, 10
Kremer J.M. et al., 2012 [17]	NCT00413660	Multi	12	7	507	53	19.9	White > 50%	Tofacitinib *−1, 3, 5, 10, 15, 20
Burmester G.R. et al., 2013 [18]	NCT00960440	Multi	12	3	399	55	16	NR	Tofacitinib *−5, 10
Kremer JM et al., 2013 [19]	NCT00856544	Multi	24	3	792	51.8	18.4	NR	Tofacitinib *−5, 10
Heijde V. et al., 2013 [20]	NCT00847613	Multi	24	3	797	52.7	14.7	NR	Tofacitinib *−5, 10
Boyle D.L. et al., 2014 [21]	NCT00976599	Multi	4	2	29	53.3	10.3	NR	Tofacitinib *−10
Tanaka Y. et al., 2015 [22]	NCT00687193	Japan	12	6	317	53.3	16.7	NR	Tofacitinib *−1, 3, 5, 10, 15
Keystone E.C. et al., 2015 [23]	NCT01185353	Multi	12	5	301	51.2	17.3	White > 50%	Baricitinib ^−1, 2, 4, 8
Tanaka Y. et al., 2016 [24]	NCT01469013	Japan	12	5	145	53.6	27	NR	Baricitinib ^−1, 2, 4, 8
NCT00902486 [25]	NCT00902486	Multi	12	4	127	55.8	19.7	NR	Baricitinib ^−4, 7, 10
Genovese M.C. et al., 2016 (a) [26]	NCT01721044	Multi	24	3	527	55.7	18	White > 50%	Baricitinib ^−2, 4
Taylor P.C. et al., 2017 [27]	NCT01710358	Multi	24	2	1307	53.3	22.7	White > 50%	Baricitinib ^−4
Dougados M. et al., 2017 [28]	NCT01721057	Multi	24	3	684	51.8	18.1	White > 50%	Baricitinib ^−2, 4
Li Z et al., 2020 [29]	NCT02265705	Multi	12	2	290	49.2	19.6	White < 50%	Baricitinib^−4
Kremer J.M. et al., 2016 [30]	NCT01960855	Multi	12	5	276	57.3	19.9	NR	Upadacitinib *−3, 6, 12, 18
Genovese M.C. et al., 2016 (b) [31]	NCT02066389	Multi	12	6	299	54.8	20.7	White < 50%	Upadacitinib *−3, 6, 12, 18, 24
Burmester G.R. et al., 2018 [32]	NCT02675426	Multi	12	3	661	55.7	21.3	White < 50%	Upadacitinib ^−15, 30
Genovese M.C. et al., 2018 [33]	NCT02706847	Multi	12	3	499	57.1	16.1	White < 50%	Upadacitinib ^−15, 30
Fleischmann R. et al., 2019 [34]	NCT02629159	Multi	12	2	1304	53.9	20.7	White < 50%	Upadacitinib ^−15
Kameda H. et al., 2020 [35]	NCT02720523	Japan	12	3	148	55.2	21.3	NR	Upadacitinib ^−7.5, 15, 30
NCT02955212 [36]	NCT02955212	Multi	12	2	338	51.7	18.9	<50	Upadacitinib ^−15
Fleischmann R.M. et al., 2015 [37]	NCT01052194	Multi	12	5	204	56.2	18.3	White < 50%	Decernotinib *−25, 50,100,150
Genovese M.C. et al., 2016 (c) [38]	NCT01754935	Multi	12	4	43	NR	NR	NR	Decernotinib ^−100, 200, 300
Genovese M.C. et al., 2016 (d) [39]	NCT2011-004419-22	Multi	24	5	358	NR	NR	NR	Decernotinib *−100^, 150, 200, 100
Takeuchi T. et al., 2016 [40]	NCT01649999	Japan	12	5	281	53	18.9	NR	Peficitinib ^−25, 50, 100, 150
Genovese M.C. et al., 2017 (a) [41]	NCT01565655	Multi	12	5	289	53.8	18	NR	Peficitinib ^−25, 50, 100, 150
Kivitz A.J. et al., 2017 [42]	NCT01554696	Multi	12	5	378	53	16.4	NR	Peficitinib ^−25, 50, 100, 150
Takeuchi T. et al., 2019 [43]	NCT02305849	Japan	12	3	519	56.7	29.7	NR	Peficitinib ^−100, 150
Tanaka Y. et al., 2019 [44]	NCT02308163	Multi	12	3	307	55.3	27.8	NR	Peficitinib ^−100, 150
Kavanaugh A. et al., 2017 [45]	NCT01894516	Multi	24	4	283	52.2	81.6	NR	Filgotinib ^−50, 100, 200
Westhovens R. et al., 2017 [46]	NCT01888874	Multi	24	7	594	53.4	19	NR	Filgotinib−50 ^, 100 ^, 200 ^, 25 *, 100 *, 200 *
Vanhoutte F. et al., 2017 [47]	NCT01384422	Republic Maldova	4	3	36	NR	NR	NR	Filgotinib−200 *, 100 ^
Vanhoutte F. et al., 2017 [47]	NCT01668641	Multi	4	3	36	NR	NR	NR	Filgotinib ^−75, 150, 300
Genovese M.C. et al., 2019 [48]	NCT02873936	Multi	24	3	449	56	19.6	NR	Filgotinib−200 ^, 100 *
Westhovens R. et al., 2021 [49]	NCT02886728	Multi	54	3	1252	53	23	NR	Filgotinib ^−100, 200

^: once in a day; *: twice in a day; NR: not reported.

**Table 2 pharmaceuticals-18-00178-t002:** Netleague table describing the ACR20 and ACR50 responses to treatment of patients with RA with various JAKinibs.

ACR50 Response							
ACR20 Response	Baricitinib	0.48 (0.17–1.33)	1.06 (0.56–2.01)	0.86 (0.44–1.69)	0.91 (0.48–1.73)	0.82 (0.46–1.45)	2.52 (1.65–3.86)
	0.68 (0.43–1.10)	Decernotinib	2.23 (0.78–6.31)	1.81 (0.63–5.24)	1.90 (0.67–5.43)	1.71 (0.63–4.68)	5.28 (2.08–13.37)
	1.14 (0.84–1.53)	1.66 (1.02–2.71)	Filgotinib	0.81 (0.40–1.63)	0.85 (0.43–1.68)	0.77 (0.42–1.41)	2.37 (1.48–3.80)
	0.89 (0.64–1.24)	1.31 (0.79–2.17)	0.79 (0.55–1.11)	Peficitinib	1.05 (0.52–2.13)	0.95 (0.50–1.80)	2.92 (1.74–4.88)
	0.89 (0.68–1.17)	1.30 (0.81–2.08)	0.78 (0.58–1.05)	0.99 (0.72–1.38)	Tofacitinib	0.90 (0.49–1.67)	2.78 (1.71–4.51)
	0.94 (0.71–1.23)	1.37 (0.85–2.20)	0.82 (0.61–1.11)	1.05 (0.76–1.45)	1.06 (0.81–1.38)	Upadacitinib	3.08 (2.10–4.51)
	1.79 (1.47–2.17)	2.61 (1.70–4.03)	1.57 (1.25–1.97)	2.00 (1.54–2.61)	2.02 (1.67–2.44)	1.91 (1.58–2.31)	Placebo

**Table 3 pharmaceuticals-18-00178-t003:** Netleague table describing the ACR70 response and HAQ-DI score for treatment of patients with RA with various JAKinibs.

HAQ-DI Score							
ACR70 Response	Baricitinib	−0.01 (−0.26–0.23)	0.04 (−0.14–0.21)	0.01 (−0.19–0.21)	−0.06 (−0.20–0.09)	0.07 (−0.09–0.22)	−0.25 (−0.36–−0.13)
	0.54 (0.17–1.73)	Decernotinib	0.05 (−0.21–0.31)	0.02 (−0.26–0.30)	−0.04 (−0.28–0.20)	0.08 (−0.17–0.33)	−0.23 (−0.45–−0.01)
	1.98 (1.47–2.66)	3.65 (1.16–11.51)	Filgotinib	−0.03 (−0.24–0.18)	−0.09 (−0.25–0.06)	0.03 (−0.14–0.20)	−0.28 (−0.41–−0.15)
	1.25 (0.73–2.14)	2.31 (0.67–7.92)	0.63 (0.38–1.05)	Peficitinib	−0.06 (−0.25–0.12)	0.06 (−0.14–0.26)	−0.25 (−0.42–−0.09)
	0.55 (0.30–1.01)	1.01 (0.29–3.59)	0.28 (0.15–0.50)	0.44 (0.21–0.92)	Tofacitinib	0.12 (−0.02–0.27)	−0.19 (−0.28–−0.10)
	0.78 (0.56–1.11)	1.45 (0.45–4.63)	0.40 (0.29–0.54)	0.63 (0.37–1.08)	1.43 (0.78–2.63)	Upadacitinib	−0.31 (−0.42–−0.20)
	3.49 (2.74–4.46)	6.46 (2.08–20.08)	1.77 (1.49–2.10)	2.80 (1.73–4.53)	6.36 (3.64–11.12)	4.45 (3.48–5.70)	Placebo

**Table 4 pharmaceuticals-18-00178-t004:** Netleague table describing ADRs and serious ADRs in treatment of patients with RA with various JAKinibs.

Serious ADRs							
ADRs	Baricitinib	0.64 (0.24–1.72)	1.00 (0.60–1.67)	0.99 (0.46–2.11)	1.14 (0.67–1.96)	0.55 (0.32–0.95)	0.91 (0.66–1.26)
	0.94 (0.58–1.52)	Decernotinib	1.56 (0.57–4.26)	1.53 (0.48–4.85)	1.78 (0.64–4.93)	0.86 (0.31–2.38)	1.42 (0.56–3.58)
	0.98 (0.69–1.41)	1.05 (0.62–1.76)	Filgotinib	0.98 (0.44–2.17)	1.14 (0.64–2.04)	0.55 (0.31–0.98)	0.91 (0.62–1.35)
	1.00 (0.71–1.42)	1.07 (0.65–1.78)	1.02 (0.69–1.51)	Peficitinib	1.16 (0.51–2.62)	0.56 (0.25–1.26)	0.93 (0.47–1.85)
	1.15 (0.86–1.54)	1.23 (0.77–1.98)	1.17 (0.83–1.65)	1.15 (0.83–1.59)	Tofacitinib	0.48 (0.26–0.89)	0.80 (0.52–1.23)
	0.76 (0.55–1.06)	0.81 (0.49–1.34)	0.78 (0.53–1.13)	0.76 (0.53–1.09)	0.66 (0.48–0.90)	Upadacitinib	1.66 (1.08–2.54)
	1.14 (0.91–1.42)	1.22 (0.79–1.88)	1.16 (0.87–1.54)	1.13 (0.87–1.48)	0.99 (0.82–1.19)	1.50 (1.16–1.92)	Placebo

ADRs: adverse drug reactions.

**Table 5 pharmaceuticals-18-00178-t005:** Netleague table describing discontinuation due to ADRs and incidence of death in treatment of patients with RA with various JAKinibs.

Discontinuation Due to ADRs							
Death	Baricitinib	0.67 (0.15–3.02)	1.00 (0.41–2.45)	1.50 (0.80–2.79)	0.82 (0.49–1.36)	0.81 (0.46–1.44)	1.10 (0.74–1.64)
	0.07 (0.00–1.55)	Decernotinib	1.49 (0.29–7.80)	2.22 (0.48–10.23)	1.21 (0.28–5.35)	1.21 (0.27–5.45)	1.63 (0.38–6.96)
	0.04 (0.00–1.55)	0.56 (0.01–20.60)	Filgotinib	1.49 (0.59–3.78)	0.81 (0.35–1.91)	0.81 (0.33–1.98)	1.09 (0.49–2.43)
	0.12 (0.00–5.65)	1.64 (0.04–75.16)	2.95 (0.04–220.61)	Peficitinib	0.55 (0.31–0.97)	0.54 (0.29–1.02)	0.74 (0.46–1.19)
	0.37 (0.03–5.46)	5.11 (0.37–71.34)	9.20 (0.34–250.95)	3.12 (0.09–07.41)	Tofacitinib	1.00 (0.59–1.67)	1.35 (0.98–1.84)
	1.09 (0.02–52.65)	15.00 (0.32–99.9)	27.00 (0.36–50.81)	9.16 (0.10–32.92)	2.93 (0.08–02.77)	Upadacitinib	1.35 (0.90–2.04)
	0.36 (0.04–3.26)	5.00 (0.59–42.07)	9.00 (0.49–166.64)	3.05 (0.13–73.37)	0.98 (0.21–4.62)	0.33 (0.01–8.17)	Placebo

ADRs: adverse drug reactions.

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
