# Peer review of "Comparative Efficacy and Safety of JAK Inhibitors in the Management of Rheumatoid Arthritis: A Network Meta-Analysis"

_pharmaceuticals, 2025, doi:10.3390/ph18020178_

Round 1
Reviewer 1 Report
Comments and Suggestions for Authors
Recommendation: Major review
In this manuscript, the author reports, " Comparative Efficacy and Safety of JAK Inhibitors in the Management of Rheumatoid Arthritis: Network Meta-Analysis ". The authors should address the following questions before getting publication.
1. Could the authors detail any additional databases besides PubMed, CENTRAL, and ClinicalTrials.gov that were considered for literature search to ensure a comprehensive review?
2. ensure the elimination of selection bias during the study selection and data extraction phases?
3. Given the variability in the quality of included studies, how did this affect the overall conclusions of the meta-analysis?
4. elaborate on the criteria used for comparing different doses of JAK inhibitors within the same drug class?
5. What confounding factors were identified, and how were they accounted for in the analysis?
6. Recommend to cite recently published research article.
DOI: 10.3390/life12122013
7. Was the Cochrane risk of bias tool adequately sensitive to assess all relevant biases in the included studies?
8. comment on the consistency of adverse events reporting across studies and its impact on the analysis?
9. Were any subgroup analyses considered to assess the efficacy and safety among different patient demographics?
10. Were sensitivity analyses conducted to assess the robustness of the findings
Comments on the Quality of English Language
Required
Author Response
Thank you very much for your great efforts we really appreciate to revise our manuscript.
Comments 1: Could the authors detail any additional databases besides PubMed, CENTRAL, and ClinicalTrials.gov that were considered for literature search to ensure a comprehensive review?
Response 1: We appreciate the reviewer’s interest in understanding the comprehensiveness of our search strategy. As detailed in our protocol (INPLASY2024100084), we conducted the literature search in PubMed, CENTRAL, and ClinicalTrials.gov, which are widely recognized and robust databases for identifying studies in biomedical research, including randomized controlled trials. These databases were selected based on their relevance to the scope of our network meta-analysis. Moreover, we performed open search and bibliographic search to ensure no key studies were missed.
Comments 2: Ensure the elimination of selection bias during the study selection and data extraction phases?
Response 2: We appreciate the reviewer’s comment regarding the elimination of selection bias during study selection and data extraction. We ensured methodological rigor to minimize selection bias through the following processes:
- Study Selection:
First-Pass Screening (FPS): Two authors independently reviewed the titles and abstracts of all retrieved records based on predefined eligibility criteria to identify potentially relevant studies.
Second-Pass Screening (SPS): The full texts of studies selected during the FPS were independently reviewed by the same two authors to assess their relevance for inclusion in the final analysis.
Conflict Resolution: Any discrepancies between the two reviewers during both FPS and SPS were resolved through discussion with a third reviewer, ensuring that decisions were objective and consensus driven.
- Data Extraction:
Data were independently extracted by two authors using standardized data extraction templates designed to capture all necessary information systematically and consistently.
Any discrepancies identified during the data extraction phase were also resolved through consultation with a third reviewer to maintain accuracy and impartiality.
By employing independent review processes and involving a third reviewer for resolving disagreements, we effectively reduced the risk of selection bias at both the study selection and data extraction stages. We believe these measures uphold the transparency and reliability of our methodology.
We trust this addresses the reviewer’s concerns and are happy to provide additional details if needed.
Comment 3: Given the variability in the quality of included studies, how did this affect the overall conclusions of the meta-analysis?
Response 3: We appreciate the reviewer’s thoughtful comment. In response, we performed a subgroup analysis comparing studies with an overall high risk of bias to those with a low risk of bias. The summary findings of this analysis, including its impact on the overall conclusions of the meta-analysis, are highlighted at the end of the discussion section.
“The subgroup analysis based on the quality of included studies highlights important differences in the efficacy and safety profiles of JAKinibs. Studies with a low risk of bias demonstrated consistent and significant efficacy across outcomes, particularly for ACR20, with manageable heterogeneity. In contrast, high-risk studies showed larger relative effects for some outcomes, such as ACR50 and ACR70, but with increased variability and heterogeneity. These findings underscore the influence of study quality on the observed magnitude and reliability of treatment effects, emphasizing the need for robust methodological designs to derive accurate and clinically meaningful conclusions”
We trust this clarification addresses the reviewer’s concern and enhances the manuscript.
Comment 4: Elaborate on the criteria used for comparing different doses of JAK inhibitors within the same drug class.
Response 4: We appreciate the reviewer’s query regarding the criteria used for dose comparison. To ensure consistency and comparability, we used the daily drug dose as the basis for comparing different doses of JAKinibs within the same drug class. This approach allowed us to standardize dosing regimens across studies and assess efficacy and safety outcomes within a uniform framework.
We updated the same in the methods section as well.
We hope this explanation addresses the reviewer’s concern and adds clarity to the methodology.
Comment 5: "What confounding factors were identified, and how were they accounted for in the analysis?"
Response: We thank the reviewer for this important observation. We identified the following potential confounding factors:
- Included population type – drug-naive/stable dose on previous therapy, inadequate response to disease-modifying antirheumatic drugs (DMARDs), inadequate response to methotrexate (MTX), and inadequate response to tumor necrosis factor inhibitors (TNFi).
- Duration of treatment – categorized as up to six weeks, 12 weeks, and 24 weeks.
To account for these factors, we performed subgroup analyses to explore differences in efficacy and safety outcomes across these variables. The findings have been summarized in a tabular format and added to the manuscript for clarity.
We hope this explanation addresses the reviewer’s concern and demonstrates the robustness of our analysis.
Comments 6: Recommend to cite recently published research article. DOI: 10.3390/life12122013
Response 6: Thank you for your suggestion to cite a recently published research article. When I accessed the DOI (10.3390/life12122013), it directed me to the article titled "Emerging Trends of Nanotechnology and Genetic Engineering in Cyanobacteria to Optimize Production for Future Applications" (available at PMC9781209). Could you kindly confirm whether this is the correct article you intended to reference?
Comments 7: [Was the Cochrane risk of bias tool adequately sensitive to assess all relevant biases in the included studies?]
Respsonse 7: Thank you for raising this important point. The Cochrane Risk of Bias (RoB) tool is widely recognized as a robust and standardized method for assessing the quality of randomized controlled trials (RCTs). In our network meta-analysis (NMA), all the included studies were RCTs, making the Cochrane RoB tool an appropriate choice for evaluating potential biases.
This tool systematically assesses a comprehensive range of biases, including:
- Selection bias (random sequence generation and allocation concealment),
- Performance bias (blinding of participants and personnel),
- Detection bias (blinding of outcome assessment),
- Attrition bias (incomplete outcome data),
- Reporting bias (selective outcome reporting), and
- Other biases (e.g., baseline imbalance, early stopping).
These domains adequately cover the spectrum of potential biases in RCTs and ensure a rigorous evaluation of study quality.
Comments 8: comment on the consistency of adverse events reporting across studies and its impact on the analysis?
Response 8: Thank you for your comment regarding the consistency of adverse events reporting across studies. The analysis of adverse events, including infection risk, herpes zoster risk, thrombotic risk malignancy risk, major adverse cardiovascular events risk, and hypercholesterolemia risk, demonstrated non-significant overall increases in adverse event risks across the included interventions. Although variability in reporting across studies was present, the safety profile of JAKinibs was consistent with the expectation of manageable risks, with some interventions (e.g., decernotinib, filgotinib) showing more favorable or less favorable risk profiles depending on the specific event. The full details of these findings have been summarized in the manuscript, with additional information provided in the supplementary tables and figures for each specific risk.
Reviewer comment 9: "Were any subgroup analyses considered to assess the efficacy and safety among different patient demographics?"
Response: Thank you for your insightful comment. We performed subgroup analyses to assess the efficacy and safety among different patient demographics and treatment contexts. These analyses were based on the following factors:
- Included population type – drug-naive/stable dose on previous therapy, inadequate response to disease-modifying antirheumatic drugs (DMARDs), inadequate response to methotrexate (MTX), and inadequate response to tumor necrosis factor inhibitors (TNFi).
- Duration of treatment – categorized as up to six weeks, 12 weeks, and 24 weeks.
- Risk of bias of included studies – high risk versus low risk.
The results of these subgroup analyses have been summarized in a tabular format and incorporated into the manuscript to highlight the observed differences.
Reviewer comment 10: "Were sensitivity analyses conducted to assess the robustness of the findings?"
Response: Thank you for the reviewer’s comment. In our analysis, we attempted to perform a sensitivity analysis (leave-one-out) to assess the robustness of the findings. However, we encountered a limitation due to the software. The leave-one-out method is not available for network meta-analysis (NMA), as we received an error message stating that the "Leave-one-out method is not available for an object of class 'netmeta'." As a result, we were unable to conduct a sensitivity analysis for the network meta-analysis.
We hope this explanation clarifies the situation.
Reviewer 2 Report
Comments and Suggestions for Authors
Dear Authors, thank your for your contribution. Comprehensive and fluid in its contruction this manuscript allow to the readers useful information about the JAKinibs already used for the treatment of RA.
Background and methodological steps using a NMA are clear. Results section, has important data including: a PRISMA flow diagram, a large table with the characteristics of the RCT selected. Very important the evaluation of the ACR response scores, including HAQ-DI score. Also, important the description about AE's effects, and mortality rates. Conclusions, support the aim for which this project was designed.
After careful review of your manuscript, I do not have any specific question or concern.
Best regards.
Author Response
Comments Reviewer 2: Dear Authors, thank you for your contribution. Comprehensive and fluid in its construction this manuscript allows the readers useful information about the JAKinibs already used for the treatment of RA.
Background and methodological steps using a NMA are clear. Results section, has important data including: a PRISMA flow diagram, a large table with the characteristics of the RCT selected. Very important the evaluation of the ACR response scores, including HAQ-DI score. Also, important the description about AE's effects, and mortality rates. Conclusions, support the aim for which this project was designed.
After careful review of your manuscript, I do not have any specific question or concern.
Response: Thank you for your positive feedback and thorough review. We appreciate your recognition of the manuscript's clarity, important data, and alignment with the project’s aim. We are grateful for your time and have no further concerns.
Reviewer 3 Report
Comments and Suggestions for Authors
In the manuscript entitled “Comparative Efficacy and Safety of JAK Inhibitors in the Management of Rheumatoid Arthritis: Network Meta-Analysis” the authors execute an NMA to evaluate the relative efficacy and safety of various JAK inhibitors and their doses in the treatment of RA. It's an exciting work due to the lack of effective therapies against RA and the high incidence of side effects of the current drugs, justifying the investment in drug discovery against this disease. However, some aspects must be observed for acceptance. Thus, my biggest criticism of the work is regarding the discussion of the results. The authors must be improved. In addition, the authors must pay attention to the following aspects:
- In the introduction, the authors must discuss better the importance of TNF-α and its inhibitors against RA and the advantages of JAKi compared with TNF-α inhibitors. For this, please read and cite the following manuscript: https://www.eurekaselect.com/article/116740
- In the introduction, the sentence “TNFi target and neutralize the activity of tumor necrosis factor-alpha (TNF-α) ……” doesn’t make sense. Please improve it.
- The authors are encouraged to add epidemiological data about RA regarding its incidence.
- The consequences of the JAK inhibitors are not evident in the introduction. The authors must improve it.
- The sentence “They have demonstrated improvements in symptoms and a reduction in disease activity for RA 67 patients ……” does not make sense. Please improve it.
- The final of the objectives must be improved. Need complementation.
- The quality of the figure 2 must be improved.
- The conclusion must be improved, and all aspects of the manuscript must be highlighted.
- The discussion must be improved. Looks like an abstract.
Comments on the Quality of English Languageimprove the English Language
Author Response
Comment 1: In the manuscript entitled “Comparative Efficacy and Safety of JAK Inhibitors in the Management of Rheumatoid Arthritis: Network Meta-Analysis” the authors execute an NMA to evaluate the relative efficacy and safety of various JAK inhibitors and their doses in the treatment of RA. It's an exciting work due to the lack of effective therapies against RA and the high incidence of side effects of the current drugs, justifying the investment in drug discovery against this disease. However, some aspects must be observed for acceptance. Thus, my biggest criticism of the work is regarding the discussion of the results. The authors must be improved.
Response 1: We appreciate your insightful comment and fully understand the concern. In light of this comment, we have revised and updated the manuscript, with significant enhancements made to the discussion section of the results.
In addition, the authors must pay attention to the following aspects:
Comment 2: In the introduction, the authors must discuss better the importance of TNF-α and its inhibitors against RA and the advantages of JAKi compared with TNF-α inhibitors. For this, please read and cite the following manuscript: https://www.eurekaselect.com/article/116740
Response 2: Thank you for your valuable comment. we have revised the introduction to include a detailed description of the role of TNF-α in RA pathogenesis and the mechanisms of TNF-α inhibitors. Additionally, we have emphasized the advantages of JAK inhibitors over TNF-α inhibitors. We have also cited the recommended manuscript (Soendergaard et al., Curr. Med. Chem., 2018) in introduction and discussion sections.
Comment 3: In the introduction, the sentence “TNFi target and neutralize the activity of tumor necrosis factor-alpha (TNF-α) ……” doesn’t make sense. Please improve it.
Response 3: Thank you for pointing out the unclear sentence. We have revised it for clarity and precision. The sentence was updated in the revised version of the manuscript
Comment 4: The authors are encouraged to add epidemiological data about RA regarding its incidence.
Response 4: Epidemiological data about RA regarding its incidence was added to the introduction section of the revised version of the manuscript.
Comment 5: The consequences of the JAK inhibitors are not evident in the introduction. The authors must improve it.
Response 5: We have revised the introduction section to clearly outline the consequences of JAK inhibitors in the revised version of the manuscript.
Comment 6: The sentence “They have demonstrated improvements in symptoms and a reduction in disease activity for RA 67 patients ……” does not make sense. Please improve it.
Response 6: The sentence was revised and updated in the revised version of the manuscript.
Comment 7: The final of the objectives must be improved. Need complementation.
Response 7: The objectives were improved in the revised version of the manuscript.
Comment 8: The quality of the figure 2 must be improved
Response 8: Thank you for the observation, we have increased the resolution of the figure 2 and inserted the same in the manuscript.
Comment 9: The conclusion must be improved, and all aspects of the manuscript must be highlighted.
Response 9: The conclusion was improved and all aspects of the manuscript were highlighted in the revised version of the manuscript.
Comment 10: The discussion must be improved. Looks like an abstract.
Response 10: The discussion was improved in the revised version of the manuscript.

Round 2
Reviewer 3 Report
Comments and Suggestions for Authors
The manuscript can be accepted. Congratulations to the authors.
Comments on the Quality of English Languagenone